# DistPar: Tensor Partitioning for Distributed Neural Network Computing

## Abstract

Existing distributed training systems suffer from the difficulties of adapting to diverse model architectures and balancing the trade-off between computational and communication costs. We introduce Distributed Partitioning (DistPar), a framework that allows users to develop parallel models with the ease of writing single-device programs. We establish the basic properties of tensor partitioning, which significantly expand the search space for optimal parallel strategies. The process of distributing global tensors from a single-device perspective is driven by the innovative use of collective communication primitives and their extensions which represent conversions between arbitrary tensor distribution properties. To further address the challenge of parallel scheme optimization, we carry out a cost function that considers both computational and communication costs. Guided by the cost function, the best-performing parallel scheme is automatically selected with configurable parameters, thus simplifying the process of developing parallel models. We demonstrate state-of-the-art results on extensive experiments. Moreover, DistPar reaches 50% higher throughput in large-scale face recognition tasks and a 20% improvement in language modeling tasks compared with data parallelism provided by PyTorch. This performance improvement aligns with the expected speedup and is particularly notable as the number of computing devices increases. The code will be released at https://github.com/DistPar.

## 1 Introduction

In recent years, deep learning has been widely applied in many fields such as image, speech, and natural language processing (Angelova et al., 2015; Ba et al., 2015; Frome et al., 2013; Gonzalez-Dominguez et al., 2015; Hinton et al., 2012; Heigold et al., 2013; Karpathy et al., 2014; Le, 2013; Maddison et al., 2015). With the increasing demand for training efficiency and data processing capabilities of deep learning, single-device training systems, although useful in certain scenarios, may struggle to meet the requirements. Hence, the distributed training approach has become an effective way to improve computing power constantly.

Distributed deep learning's performance relies primarily on efficient collective communication to adapt to different given computational devices (Yuan et al., 2022; Lepikhin et al., 2020). Existing deep learning parallelism libraries have made great efforts on it. Typically, parallelization strategies in the context of distributed deep learning include two main aspects: data parallelism and model parallelism. Data parallelism, the former, entails the further subdivision of a mini-batch of data, subsequently distributed across computational nodes, which facilitates the training of substantial volumes of data (Baruah et al., 2022; Shallue et al., 2018; Nguyen & Wahib, 2021; Herlihy et al., 2021; Krizhevsky, 2014). Model parallelism, the latter, is conventionally applied to partition neural networks into segments that are subsequently deployed across computational nodes (Dean et al., 2012; Narayanan et al., 2021; Huang et al., 2018; Harlap et al., 2018; Shoeybi et al., 2020; Xu et al., 2021; Wang et al., 2021; Bian et al., 2021a). Based on the parallelism strategies mentioned, we believe a comprehensive approach that aggregates them with each other, enables faster computation and efficient utilization of computational devices.

Existing parallelism libraries like Pytorch, its DistributedDataParallel interface is challenging to users, because it requires users to design the communicative module of parallelism strategies manually. Hence, it's necessary for us to design a set of parallel operation semantics from the bottom to

achieve an end-to-end structure so that users can handle parallel training tasks on multiple devices with the same ease as a single device.

Our unified strategy, DistPar, introduces a set of tensor partitioning attributes aimed at instructing the allocation of global logical tensors to specific physical devices—referred to as physical tensors for simplicity. DistPar merges these devices into a coherent logical supercomputer, allowing developers to handle parallel training tasks on multiple devices as simply as a single device. This enhanced accessibility for individual users, so they can focus on more top-level design.

The process of distributing global tensors from a single-device perspective is driven by the innovative use of collective communication primitives and their extensions which represent conversions between arbitrary tensor distribution properties. This capability is integrated into DistPar through the inclusion of pass layers. Therefore, DistPar effectively enhances the extensibility, enabling to be adaptive to different model structure and computational device.

To further address the challenge of parallel scheme optimization, DistPar assesses the cost in a comprehensive manner, which combines the conversion of parallel attributes across various parallelization strategies. At the meantime, to simplify the process of designing and selecting the best scheme, we provide a configurable parameter so that users can easily optimize computational cost and communication cost collaboratively and automatically. Evidently, the cost design helps users to adapt to different computational devices and design their own parallelism program easily.

**The overall contributions are as follows**:
• We present a novel tensor partitioning strategy, DistPar, aimed at generating a comprehensive range of parallelization strategies.
• We employ meticulously designed intermediate primitives to facilitate the automatic transformation of distributed properties within the context of physical tensors. These mechanisms naturally support arbitrary parallelization combinations.
• We introduce cost hyperparameter to generate different parallelization strategies, enabling the user to evolve the selection of optimal parallelization schemes.
• We prove that DistPar attains state-of-the-art performance in standard benchmark assessments.

## 2    RELATED WORKS

Numerous distributed parallelism strategies exist, with data parallelism and model parallelism being as the most widely adopted approaches.

**Data parallelism** involves dividing a mini-batch of data into smaller segments and distributing them to different computational nodes (Baruah et al., 2022; Shallue et al., 2018; Nguyen & Wahib, 2021; Herlihy et al., 2021; Krizhevsky, 2014). In data parallelism (Krizhevsky, 2014), each device retains a complete copy of the distributed neural network (DNN) model and processes a portion of the entire training dataset. This approach enables the training of large datasets, thereby enhancing both the scale and speed of training. However, data parallelism introduces inter-device communication overhead during the synchronization process when model weights are updated. This issue can become more apparent as the model size increases, which poses some challenges to the scalability and compatibility of data parallelism.

**Model parallelism** offers an alternative to data parallelism by directly partitioning DNN models across devices. With model parallelism (Kingma & Ba, 2017; Fang et al., 2023), weight parameters within the model are distributed among the available workers, which are typically GPUs. This approach consists of two main components: tensor parallelism and pipeline parallelism.

**Tensor parallelism** involves splitting tensors across an array of devices, typically occurring between the forward and backward propagation phases (Shoeybi et al., 2020; Xu et al., 2021; Wang et al., 2021; Bian et al., 2021a; Wang et al., 2021; Bian et al., 2021b; Cannon, 1969; Berntsen, 1989; van de Geijn & Watts, 1995; Solomonik & Demmel, 2011). Megatron-LM(Shoeybi et al., 2020) introduced 1D tensor parallelism, which divides the linear layer along either the column or row dimensions. When employing tensor parallelism, communication tends to be frequent, and the data volume transferred during these communications is often substantial.

**Pipeline parallelism** divides the model on a layer basis, occurring at the junction of adjacent stages (Huang et al., 2018; Harlap et al., 2018; Li & Hoefler, 2021). Recent developments, such

as GPipe(Huang et al., 2018), have introduced pipeline parallelism, which involves synchronous weight updates. In this case, communication remains frequent but typically involves smaller data volumes. Due to the inherent characteristics of pipeline parallelism, amounts of device idle time called bubbles are generated.

**Comparison**. To reduce communication volume, tensor parallelism is preferred. Meanwhile, to improve peer-to-peer communication, pipeline parallelism is a suitable choice. However, it is equally important to note that bubbles cost a significant amount of time. To mitigate this, it is recommended to limit the number of pipeline stages to the number of micro-batches. In practice, when the level of tensor parallelism matches the number of devices, performance tends to reach its peak.

Other optimized strategies, as demonstrated in previous studies (Jia et al., 2018a;b), concentrate on tensor-related refinements along multiple axes to determine the most optimal parallelization strategy.

Achieving high throughput at a large scale demands innovative and intricate design across various facets. This includes the intelligent partitioning of computational graphs onto devices to minimize data transfer over the network while minimizing device idle time. It also involves the implementation of communication optimizations specific to the domain.

**Unified strategy**. Based on the comparisons mentioned earlier, we conclude there is an imperative need for a unified strategy that amalgamates various advantages. A commonality observed in existing parallelization strategies is the shared goal of optimizing the utilization of computational resources and enhancing overall computational efficiency. However, it is crucial to acknowledge that a single parallelization strategy often struggles to meet the efficiency requirements of complex business models. These individual parallelization strategies fall short in planning and executing the global logical computational graphs effectively. Therefore, a holistic approach to the entire process is necessary. We have identified three key indicators—accessibility, compatibility, and communication cost—as crucial elements to facilitate comprehensive considerations.

## 3 METHODOLOGY

This section establishes the theoretical foundation for subsequent experiments detailed in Section 4. We also introduce the proposed intermediate primitives designed to optimize model communication cost. Moreover, we illustrate complex operations using intermediate primitives. To be clear, we induce the transformations of distributed properties, offering a comprehensive perspective on distributed computation and collective communication. Finally, we employ partition analysis to quantitatively assess associated expenses in the theory.

### 3.1 DISTRIBUTED PROPERTIES

Many parallelism strategies suffer from the bottleneck to be adaptive to different model structures and computational devices, so we need to design parallelism operation semantics from the bottom of the distributed training system. In this way, we can satisfy arbitrary parallelism strategies and their extensions. Distributed properties involve various parallel-related terms, with the goal of modeling global distributed computation by parameterizing operator deployment schemes. Within the modeling framework, developers have access to flexibly construct algorithmic models and configure distributed attributes according to their preferences. Formally, distributed properties are defined as a set of parameters associated with primitive operators. Their core framework involves the registration of operators along with their distributed attribute signatures. Here, we define the framework and further explain it with a qualitative analysis. Specifically, we discuss four key distributed properties: Placement, Scatter, Broadcast, and PartialReduce.

**Placement** of each operator in the logical graph specifies the devices where logical operators will be deployed. In the case of common data parallelism, all operators are deployed to all devices. Logically, all operators are designed to run on a single device, but in practice, they operate on different devices based on their placement configuration.

**Broadcast** is a procedure that involves sending the complete data of a logical tensor to all other computational nodes in the cluster, resulting in the creation of physical tensors that are copies of the logical tensors. Its process ensures that each physical operator has access to the entire dataset stored in the logical tensor. For convenience, we denote the Broadcast attribute as B.

**Scatter** involves splitting data from a logical tensor into chunks and sending these chunks to devices in a certain order. This creates local physical tensors. The Scatter property is characterized by a single parameter for partitioning, denoted as S(0) for horizontal slicing and S(1) for vertical-axis slicing. Scatter represents a one-to-multiple distribution similar to Broadcast. Their distinction is that Broadcast sends identical copies to all devices, whereas Scatter sends different chunks to each device. For simplicity, we denote Scatter as S.

**PartialReduce** signifies that the physical and logical tensors have matching shapes, but the values in the physical tensors constitute a subset of those in the logical tensors. Figure 1(a) illustrates the characteristics of PartialReduce. The complete global logical tensor can be reconstructed by reducing the physical tensor at the target location across all devices. Logically, the global logical tensor Y is obtained by the logical tensors U and V. However, in the physical implementation, component $U_0$ of logical tensor U, sliced by S(1), and component $V_0$ of logical tensor V, with S(0), are deployed on device 0. They are utilized to execute the corresponding operator, yielding the local physical tensor $Y_0$. Meanwhile, we use the same operation to obtain $Y_1$. Consequently, Y can be reconstructed by reducing $Y_0$ and $Y_1$. Furthermore, $Y_0$, $Y_1$, and Y share an identical shape.

### 3.2 CONVERSIONS OF DISTRIBUTED PROPERTIES

This section derives the intermediate primitives and their variants, such as complex operation construction, and conversions between distributed properties, and also mentions the crucial intermediate primitives for converting diverse distributed attributes and evaluating the associated communication cost. The optimal parallel strategy selection relies on minimizing communication overhead. Converting tensor distributed attributes between devices incurs overhead, except when executed on the same device, in $S2P$, which eliminates communication costs. However, cross-device communication cost in conversions is proportional to the size of the logical tensor T. Furthermore, induced from the modeling, we introduce existing intermediate primitives. The combinations of primitives and various conversions between distributed properties have been shown in Appendix A.1, and the complex operations are included in Appendix A.2.

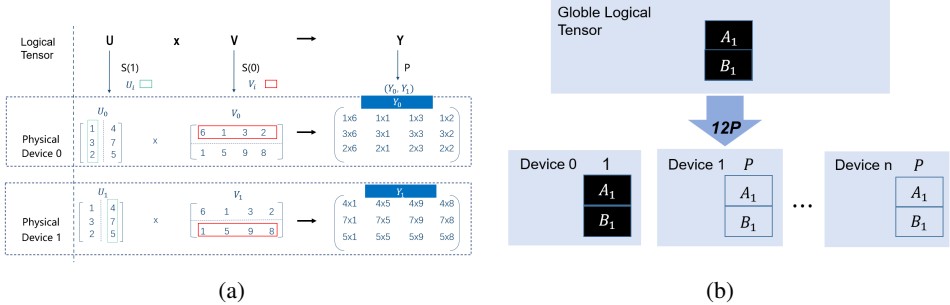

(a)                                         (b)

Figure 1: An example of a PartialReduce procedure(a), where PartialReduce is denoted as P, and the behavior of $12P$(b), $12P$ is an atomic operation deploying a global logic tensor to a local reduction, where one device places a physical tensor, a copy of the global logic tensor, other devices only place physical tensors that have the same shape as the global logic tensor but with all values set to zero.

### 3.3 IMMEDIATE INFERENCE

Immediate inference involves deducing the distributed properties of the output from the attributes of the input tensor. Table 1 in Appendix A.1 illustrates the process of directly inferrable distribution using the matmul operator, where each case of the input's properties is specified, and the valid output's distributed properties are inferred. It takes a global logical tensor as input and infers the distributed attributes of local physical tensors across all devices. If the inference depends on the assistance of intermediate primitives, we select the most cost-effective primitive to insert between the input and the local physical tensor beforehand. When two adjacent operators establish a producer-consumer relationship and the distributed properties of the output tensor from the producer operator do not align with the properties required by the consumer operator, DistPar needs to dynamically derive intermediate transformation primitives. These primitives are automatically inserted between

the producer and consumer operators through the pass layers to ensure alignment. We present an example of inferring the intermediate primitive AllGather in Appendix A.1.2

### 3.4 COST DESIGN

The overall cost is evaluated based on both computational cost and communication cost. To be specific, in order to optimize computational cost and communication cost collaboratively, we need to characterize the trade-off between them. Therefore, we introduce the ratio of computational cost to communication cost, which is denoted by beta.

**Computational Cost** in DistPar is simplified to the sum of the elements of the input and output tensors corresponding to different parallelization strategies, due to the fact that DistPar assumes all parallelization strategies use the same operator library.

**Communication Cost** is defined as the total communications across multiple devices. In our implementation, communication cost is estimated using the conversion cost that results from the conversions of distributed properties. Details are revealed in Appendix A.1.

## 4 EXPERIMENTS

In this section, we conduct a comparative analysis of DistPar, TensorFlow, and Pytorch to demonstrate the effectiveness of DistPar.

### 4.1 SYSTEM PERFORMANCE

**Setup**. We conducted a comparative evaluation, analyzing ResNet-50 pre-trained on the ImageNet-2012 dataset (Heigold et al., 2013) for image recognition and the BERT-Base model (Karpathy et al., 2014) for query answering in natural language processing tasks. We assessed the throughput and speedup of these models implemented with DistPar, as well as the data parallelism libraries of PyTorch and TensorFlow. It is worth noting that our emphasis is on system performance metrics rather than learning objectives.

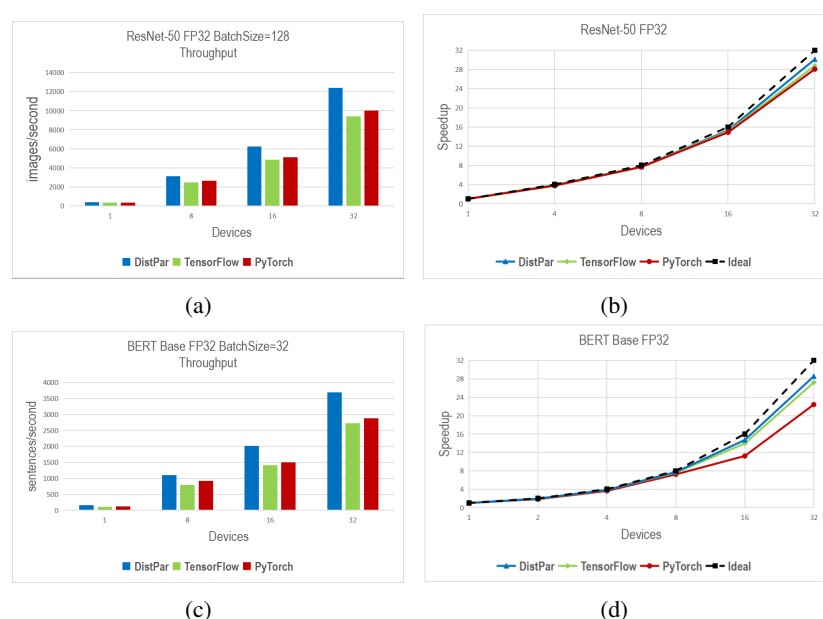

Figure 2: Training speed for 2 models using 32-bit floats. Throughput is measured in images per second for the ResNet-50 and in sentences per second for the BERT Base model. The fastest speed for each model is shown in the group of green rectangles in subplots (a) and (c). Larger batch sizes narrow the distance between DistPar's speedup curve and the ideal curve, indicating that DistPar can effectively leverage system scalability with large-scale datasets in subplots (b) and (d).

**Analysis**. We analyze the system performance in view of throughput and speedup. On mainstream models for various tasks, namely ResNet-50 in Figure 2(a)(b) and BERT in Figure 2(c)(d), we conducted a comparative evaluation on the performance of DistPar's automatically selected parallelism strategy against data parallelism in PyTorch and TensorFlow frameworks.

•**Throughput Comparison** Figure 2(a) and (c) illustrate the variation in the throughput performance of the three libraries as the number of computational devices changes. When comparing the throughput of DistPar-implemented ResNet-50 models with 16 and 32 computational devices, it is observed that they outperform the suboptimal PyTorch implementation by 1500 and 2300 images/second, respectively. In the case of BERT-base models, the respective throughput improvements are 500 and 750 sentences/second. As depicted in Figure 2(a) and (c), which illustrate the throughput of DistPar across various numbers of computational devices, it's evident that DistPar consistently outperforms the comparative frameworks. Furthermore, this advantage becomes more obvious as the scale of computational devices increases. These findings underscore the superior overall throughput performance of DistPar, owing to its designed and selected global parallelization strategy in comparison to the data parallelism strategy employed by the comparative frameworks.

•**Speedup Comparison** Figure 2(b) and (d) illustrate the variation in the speedup performance of the three libraries as the number of computational devices changes. With the increase in the number of devices, it becomes more evident that both the ResNet-50 model(b) and the BERT model(d) implemented with DistPar(blue curve) closely approach the ideal system(black curve), while TensorFlow (green curve) follows DistPar as the next best option. For ResNet-50 model(b) and BERT model(d), when the number of computational devices reaches 32, they achieve speedups 2 and 5 times higher than PyTorch(red curve), respectively. This indicates that when dealing with a larger number of computational devices, the performance improvement of DistPar over PyTorch's data parallelism strategy becomes more notable. These results collectively highlight that, in comparison to the baselines, DistPar exhibits enhanced system scalability. From the figure, it's clear that DistPar outperforms the existing TensorFlow and PyTorch. When batch sizes get larger, the distance between DistPar's speedup curve and the ideal curve is narrowed, indicating that DistPar can effectively leverage system scalability with large-scale datasets, showcasing its promising adaptability. In summary, DistPar can boost the system's overall performance including throughput and speedup, and achieve promising results compared with popular deep learning parallelism libraries.

## 4.2 HYPERPARAMETER OPTIMIZATION

**Setup**. This experiment demonstrates DistPar's optimization of parallelization strategies, as Figure 3 shows. The definition of overall cost can be found in Section 3.4. Specifically, the evaluating environment is configured with 4 * NVIDIA GeForce GTX 1080 GPU.

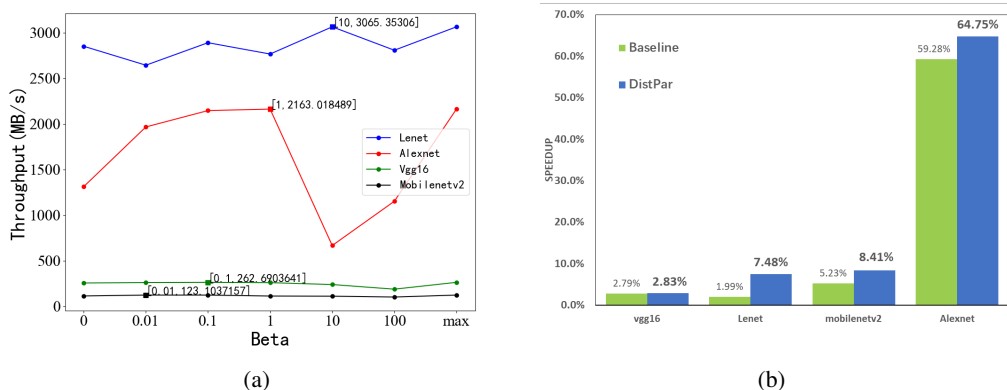

(a)                              (b)

Figure 3: Results of the hyperparameter optimization experiment. Since the values of beta corresponding to the maximum throughput vary on different models, we can select the optimal parallelism strategy for each model by adjusting the value of beta (a). Compared with the cost design of baselines that only takes communication cost into account, DistPar has notably better performance due to its collaborative optimization on both computational cost and communication cost (b).

**Analysis**. DistPar exhibits varying parallelization strategies based on the ratio of computational cost to communication cost, denoted as the hyperparameter beta. This leads to different distribution characteristics of input and output tensors for the operators comprising the model. For different models, the beta value corresponding to the maximum throughput varies. For LeNet, AlexNet, Vgg16, and MobileNetV2, the beta values corresponding to their respective maximum throughputs are 10, 1, 0.1, and 0.01, with the corresponding speedup percentages being 7.48%, 64.75%, 2.83%, and 8.41%. The results highlight that DistPar adapts its parallelization strategy based on beta, resulting in different throughput outcomes. It is worth noting that the beta value corresponding to the maximum throughput is not consistent with the baseline which only considers the communication cost. This implies that, compared to a baseline approach that only considers communication cost, DistPar effectively leverages both computational and communication costs to guide its parallelization strategy selection. In summary, DistPar empowers users to optimize parallelization strategies for different models by fine-tuning the hyperparameter beta. This enables the selection of the parallelization strategy that corresponds to the maximum throughput for each model.

## 4.3 SCALABILITY ANALYSIS

**Setup**. In order to observe the DistPar's implementation of the large-scale face recognition insightface model, we conduct a series of separate experiments. The throughput on the insightface model was evaluated on different batch sizes and the number of categories. The configured with 8 GPUs of NVIDIA Tesla V100, FP32. Moreover, data parallelization with Broadcast and model parallelization with S1. To explore more cases, we vary the batch size and parallelization options for the fully connected layer of the last layer of the insightface model. As shown in Figure 4.

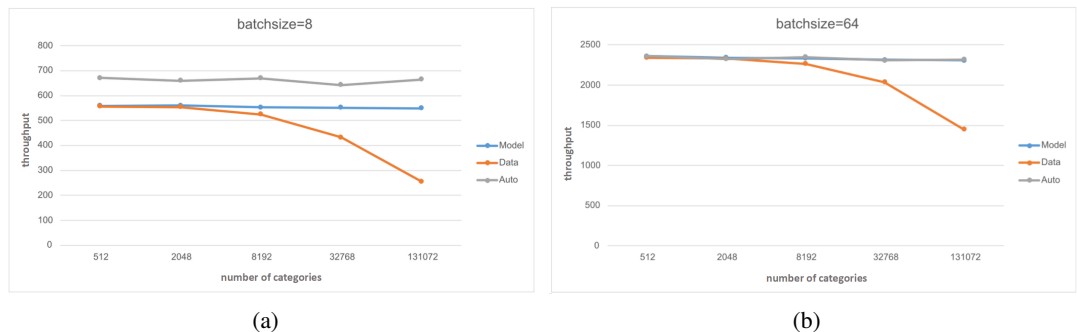

(a)                                                                                           (b)

Figure 4: Performances of DistPar, data parallelization, and model parallelization, with batch_size fixed to 8 and 64. As the number of categories and the batch size vary, DistPar shows an identical pattern of prioritizing data parallelism when the number of categories is small and tends to select model parallelism when it is gradually increasing. DistPar can outperform data parallelism by 120% and 50% within batchsize fixed to 8 and 64 respectively, which confirms that DistPar is able to automatically plan and select the better parallelization scheme that is adaptive to different computational resources according to different tasks.

**Analysis**. Based on the Insightface model structure for face recognition tasks, we analyze the impact of changes in the number of categories on the selection of DistPar parallelization strategies. When the number of categories is small, data parallelism performs similarly to model parallelism and maintains a relatively good performance. However, as the number of categories increases, the throughput of data parallelism decreases. On the other hand, the performance of the model parallelism strategy remains stable. For DistPar, when the number of categories is low, it favors data parallelism. However, as the number of categories increases, DistPar tends to choose model parallelism as the overall strategy. These experimental results confirm that DistPar has the capability to select the optimal parallelization strategy that matches different numbers of categories effectively. Furthermore, we analyze the impact of batch size on the selection of DistPar parallelization strategies. When the batch size is small, DistPar exhibits better compared to data parallelism and model parallelism. As the batch size increases, the performance of DistPar remains competitive with model parallelism. It's worth noting that when the batch size is 128, DistPar's performance is slightly lower than that of model parallelism. However, by adjusting the hyperparameter beta, DistPar can be fine-tuned to

match the performance of model parallelism. These experimental results confirm that DistPar can adapt to different batch sizes and select the optimal parallelization strategy accordingly.

## 4.4 OPTIMIZATION SPACE

**Setup**. We conducted comparative experiments on the last three fully connected layers of the VGG16 network using DistPar with the manual configuration strategy provided by PyTorch, which involves a potential combination of all parallel strategies, DistPar implements an optimal parallelization strategy suitable for the last three layers.

**Analysis** From the experiments, the throughput of the data parallelism strategy DDD configured in PyTorch is the lowest, as shown in Figure 5. By introducing some degree of model parallelism, the overall performance of VGG16 is improved. Considering the large dimension of the first fully connected layer, configuring it with the S0 parallelization strategy yields favorable results. The results indicate that the manually configured optimal parallelization strategy in PyTorch is RCR, confirming that the S0 parallelization strategy is best suited for the first fully connected layer.

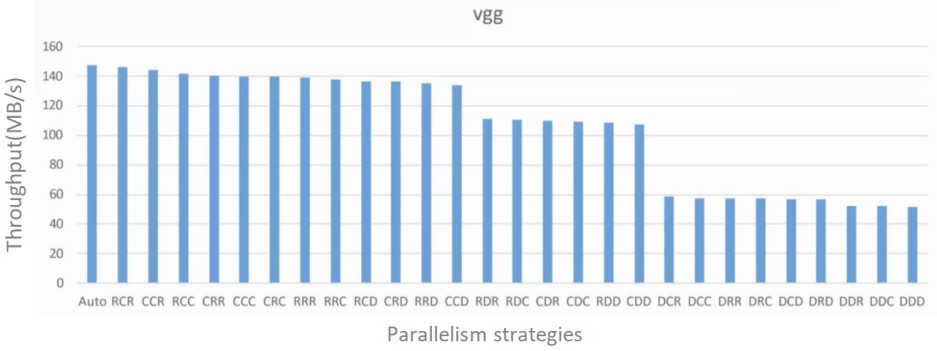

Figure 5: Performance evaluation of all possible parallelism strategies. where "Auto" describes the DistPar strategy, "R" represents S0 parallelism, "C" represents S1 parallelism, and "D" represents data parallelism. Specifically, the PyTorch configuration using the RCR parallel strategy, as illustrated in the figure, describes the optimal setup: the first fully connected layer employs S0 parallelism, the second layer utilizes S1 parallelism, and the third layer again adopts S0 parallelism.

Compared to the manually configured PyTorch parallelization strategy, the DistPar strategy exhibits significant performance improvements. In PyTorch's manual configuration approach, only the distributed attributes affecting variable operations are determined, while the parallelization strategy for intermediate tensors remains undetermined. Meanwhile, DistPar has the capability to comprehensively select and optimize parallelization strategies for intermediate tensors, analyzing operators within the backward computation graph to determine the best parallelization strategy. In contrast to Pytorch's manual configuration approach, DistPar has a larger search space. In summary, compared to manually configured PyTorch parallelization strategies, DistPar yields superior performance, resulting from DistPar's larger search space and its optimization capabilities.

## 4.5 PRIMITIVE-LEVEL OPTIMIZATION

**Setup**. DistPar offers multiple implementations for the same parallelization strategy. For example, as shown in Figure 3(b) (see Appendix A.3), the S2B transformation can be realized using both the AllGather approach and a combination of Gather and Broadcast. In order to investigate how DistPar's use of different implementations for the same parallelization strategy affects system throughput performance, we evaluated the throughput performance of various collective communication operations, including ReduceScatter, AllGather, and AllReduce, as they vary with the scale of computational devices, using the Enflame-CloudBlazer T10-16GB DCU in the same environment.

**Analysis**. In Figure 6, the results indicate that different communication primitives exhibit various throughput performances at the same number of computational devices. The overall throughput trends for all primitives show a pattern of initial decline followed by stabilization as the scale of

computational devices increases. When there are 8 devices, the throughput of AllGather is 10.36 and 12.40 times higher than ReduceScatter and AllReduce, respectively. This suggests that when the number of computational devices is relatively low, significant performance differences exist among different communication primitives. As the number of devices increases to 320, these differences are reduced to 1.03 and 1.0, respectively, indicating that the performance gap between different primitives gradually narrows with the growth in the number of computational devices. This experiment confirms that, when the number of computational devices is low, DistPar exhibits significant performance variations based on different communication primitives, expanding the candidate space for selecting the optimal strategy for the same parallelization strategy. When the number of computational devices is high, DistPar's implementations based on different communication primitives for the same parallelization strategy tend to have stabilized performance differences, highlighting DistPar's ability to select the most stable and highest throughput implementation when there are a significant number of computational devices.

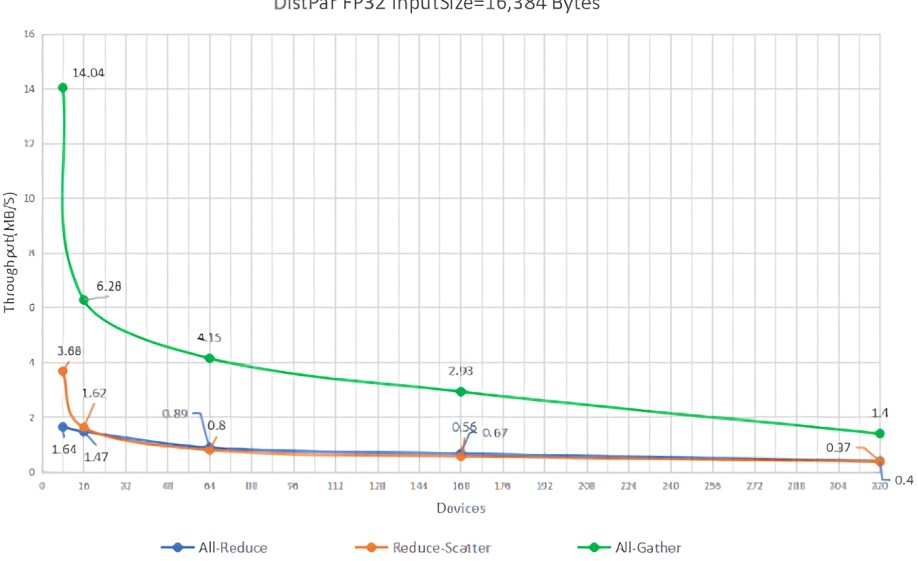

Figure 6: The throughputs for data parallelism with different tensor partition options in DistPar. This figure illustrates throughputs of varied intermediate primitives are different under the same device. Notably, throughputs for all primitives initially drop before plateauing. This decline is due to the reduced communication bandwidth between devices as the parallel width of collective communication widens, leading to less bandwidth utilization by individual intermediate primitives.

## 5   CONCLUSIONS AND FUTURE WORK

In this paper, we propose DistPar, a unified approach for efficient tensor partitioning in parallel computation of neural networks, and describe the methodology of determining solution spaces for attribute conversions in distributed training systems. The results indicate that the proposed tensor partitioning approach of DistPar supports flexible combinations of various parallelism strategies. Furthermore, under the collaborative guidance of computational cost and communication cost, DistPar enables users to select the parallelism strategy that yields the maximum throughput corresponding to different models. Hence, we believe DistPar is very promising in related domains. However, there are potential limitations that need to be considered. We qualitatively discuss the relationship between cluster communication performance and parallel width. As the parallel width $n$ of collective communication increases and the input data size $|T|$ remains constant, both the total communication volume across devices and the memory savings on each device grow proportionally. The time required for a specific collective communication is not affected by the parallel width $n$. Consequently, as $n$ increases, DistPar can utilize a bandwidth of size $(n-1) \times |T|$ for inter-device communication. This benefits in two ways: Firstly, each device can process a smaller data portion, $\frac{|T|}{n}$, leading to faster computation; Secondly, memory savings increase by $(n-1) \times |T|$, thus future work needs to build the model of communication efficiency and communication bandwidth through experimental simulation.

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
