# DistPar: Tensor Partitioning for Distributed Neural Network Computing

## A  Appendix

### A.1  Immediate Inference

#### A.1.1  Extensions

This section shows the extensions of existing intermediate primitives in order to effectively introduce new primitives or combinations of existing atomic operations, hence broadening the family of intermediate primitives.

**Atomic Operations**. We define a series of atomic operations as the minimum unit to implement a complex intermediate primitive.

**All2All** converts Scatter attribute $S_i$ to Scatter attribute $S_j$, based on the inference results of $S_i 2 S_j$ in Figure 3(a).

**AllGather** defines the transition of Scatter property $S$ to Broadcast property $B$, following the conclusion of the inference of $S2B$ in Figure 3(b).

**ReduceScatter** converts PartialReduce property $P$ to Scatter property $S$ considering the results for $P2S$ obtained in Figure 3(c).

**AllReduce** converts PartialReduce property $P$ to Broadcast property $B$.

**Identity** defines the conversion of a distributed attribute to the same attribute. Since distributed attributes of the input and output tensor are of the same, Identity primitive is a copy operation that represents the copying process from the input tensor to the output tensor.

**12P** is introduced as an atomic operation that deploys a global logic tensor to a local reduction, where one device places a physical tensor, a copy of the global logic tensor, while other devices only place physical tensors that have the same shape as the global logic tensor but with all values set to zero.

#### A.1.2  Partition Analysis

Efficiently implementing a distributed neural network hinges on maximizing the utilization of bandwidth between devices and equipment. To illustrate how DistPar optimally leverages bandwidth, we examine the atomic operations, Reduction-Scatter, detailed in Appendix A.3, as a case study. We implement Reduction-Scatter and analyze how this operation effectively harnesses the available bandwidth between devices.

**Basic Settings**. Assuming duplex bandwidth of $\beta$ per device and a cumulative egress bandwidth of $n \times \beta$ for all devices, the global logical tensor with size $|T|$ is divided into $n$ copies on each device, resulting in each inter-device communication involving data of size $\frac{|T|}{n}$. In ring-based Reduce-Scatter, each device communicates with the others a total of $n - 1$ times.

**Memory Saving**. Given that the data size processed by Reduce-Scatter on each device is $\frac{|T|}{n}$, each device can conserve $\frac{(n-1)|T|}{n}$ of memory space. On a cluster with $n$ devices, this translates to a collective memory savings of $(n - 1) \times |T|$.

**Bandwidth Occupancy**. Throughout the process, each device exchanges data of size $\frac{(n-1)|T|}{n}$ with other devices, resulting in a total data transfer size of $(n - 1) \times |T|$. This data transfer is directly

Table 1: Tensor partitioning of matmul with immediate inference. It is obvious if the value of either input is $B$, then $Y$ equals to another input. To be specific, when the input X exhibits S(0) and the input weight tensor W exhibits B, the distributed property of the output is determined to be S(0), indicating that the operator is performing data parallelism. Conversely, when X has B and the input weight tensor has S(1), the distributed property of the output is inferred to be S(1), signifying that the operator is implementing model parallelism.

| Tensor | Input X | Input W | Y=XW |
|---|---|---|---|
| | S(0) | B | S(0) |
| | B | S(1) | S(1) |
| **Distributed** | S(1) | S(0) | P |
| **Property** | P | B | P |
| | B | P | P |
| | B | B | B |

proportional to the number of devices, $n$. Importantly, both the ingress and egress bandwidths of each device are fully utilized, with no contention for bandwidth.

**Time Consumption**. Given that each device has an ingress or egress bandwidth of $\beta$, and the communication data generated by each device is of size $\frac{(n-1)|T|}{n}$, the time required for the entire process is $\frac{(n-1)|T|}{n \times \beta}$. Notably, as the number of devices, $n$, increases, the total communication time remains approximately constant at $\frac{|T|}{\beta}$. Remarkably, the time required is independent of the number of devices, $n$.

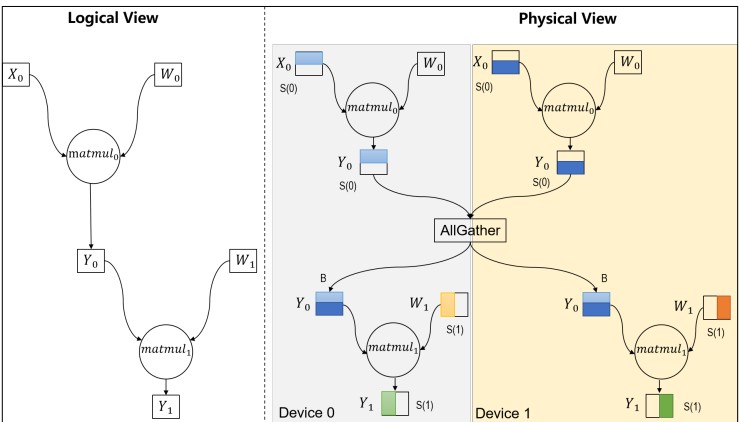

Figure 1: An example of inferring the intermediate primitive AllGather. Specifically, the requirement for B by the consumer operator $matmul_1$ necessitates the aggregation of data from the output tensor of the producer operator $matmul_0$ on all devices. The AllGather communication primitive accomplishes it, resulting in its derivation as the appropriate operator.

## A.2 COMPLEX OPERATION CONSTRUCTION

**Division Expression**. Introducing an intermediate primitive is feasible when it comes to converting two distributed attributes in one step. Therefore, we give a division expression that inserts an intermediate primitive between the source operation and the target operation. We decompose the conversions from the source operation to the target operation into a chain of processes, involving intermediate primitives. The Division Expression (1) is as follows:

$$Target\ I2O = Division(Atomic\ I2M, Atomic\ M2O) \tag{1}$$

The target operation $I2O$ can concatenate two atomic operations $I2M$ and $M2O$ via the intermediate distributed property, called $M$. When the distributed attributes $I$ of the input, $O$ of the output

tensor, and the intermediate attribute $M$ satisfy the conversion conditions of the atomic operations $I2M$ and $M2O$, the target distributed attribute derived from the target operation–$I2O$ transformation can be obtained. A division expression is a concatenation of multiple atomic operations via intermediate distributed properties. Figure 2(a) illustrates the construction of a complex operation $12S$ using a division expression.

**"OR" Expression**. We introduce the "or" expression to construct complex intermediate primitives. We define the "or" Expression (2) as follows:

$$Target\ X2W = (Atomic\ I2W)\ |\ (Atomic\ O2W) \tag{2}$$

The "or" expression indicates that the derivation of the target operation $X2W$ can be satisfied, if it occurs that the distributed properties of the input and output tensor satisfy the derivation of either the distributed derivations of atomic operation $I2W$ or that of $O2W$. Where $X$ stands for arbitrary distributed property, $W$ for intermediate distributed property, $I$ for the distributed property of the input tensor, $O$ for the distributed property of the output tensor, and 2 implies $S2B$ conversions.

The target operation resulting from an "or" expression implies the parallel combination of multiple atomic operations. Figure 2(b) illustrates the process of constructing a target operation $X2B$ using an "or" expression, where the distributed attribute $X$ denotes an arbitrary distributed attribute. The process of constructing a target operation $X2B$ for the conversion of an arbitrary distributed attribute $X$ into a broadcast attribute $B$: The derivation of the distributed attribute of the target operation $X2B$ is satisfied as long as the derivation of the distributed attribute of any of the following atomic operations $B2B$, $P2B$ and $S2B$ is satisfied.

**Compositions Extensions**. To represent individual complex target operations, we can combine "or" expression and division expression in various ways using compositions. Figure 2(c) illustrates the process of constructing a complex operation $12X$ using a composite approach.

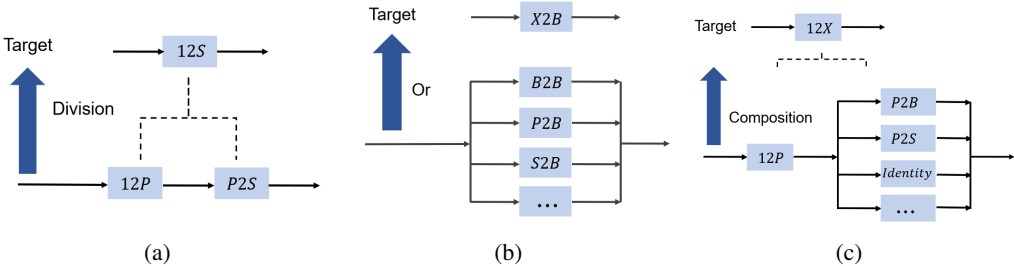

Figure 2: The behaviours of division expression(a), "or" expression(b) and composition extension(c)

### A.3 CONVERSIONS OF DISTRIBUTED PROPERTIES

$S_i2S_j$ presents the intermediate primitives to pair conversions of $S_i$ and $S_j$. According to Figure 3(a), A tensor with property S(0) consisting of data $A_1$ and $B_1$ is on device 0, and $B_1$ is empty. Device 1 places a tensor with property $S(0)$ consisting of data $A_2$ and $B_2$ with $A_2$ empty. The tensor with Scatter property $S_1$ on device 0 is from sending $A_1$ and $A_2$ to device 0. Figure 3(a) shows the derivation of the primitives to the conversion. Similarly, the tensor with Scatter S(1) on device 1 is from sending $B_1$ and $B_2$ to device 1. The sender and receiver buffers of each device are arrays partitioned into a few data blocks. Here, the All2All primitive is to send the i-th block of the sender buffer of all devices to the receiver buffer of device i, thus to help $S_i2S_j$ conversion. The communication overhead is the size of the global logical tensor $|T|$.

$S2B$ illustrates the intermediate primitives' inferences when Scatter attribute S is converted to Broadcast attribute B, as shown in Figure 3(b). $A_1$ block with attribute $S(0)$ on device 0 and the $B_2$ block with attribute $S(0)$ on device 1 are sent to the receiver buffer of device 1. We use the primitive AllGather to broadcast the aggregated tensor the receiver buffers of all output devices, this step can be divided into Gather and Broadcast as shown in Figure 3(b). Communication overheads of $S2B$ conversions is $n*|T|$, the product of the number of devices placing output tensors $n$ and the size of global logical tensor $|T|$.

***P2S*** infers the intermediate primitives for the conversion of PartialReduce attribute P to Scatter attribute S, as shown in Figure 3(c). The data on device 0 and device 1 is reduced. The result is flowed to the receiver buffer of the specified device. Then, specific data is divided and distributed to device 0 and device 1. We use the primitive ReduceScatter to achieve it, which can be partitioned into PartialReduce and Scatter, as shown in Figure 3(c). Communication overheads for ***P2S*** conversion is $n * |T|$, the product of the number of devices placing the input tensor $n$ and the size of the global logical tensor $|T|$.

***P2B*** indicates the inferring of the intermediate primitives for the conversion of PartialReduce attribute P to Broadcast attribute B, as shown in Figure 3(d). The data on device 0 and device 1 is subjected to a reduction operation and then the result is written to the receiver buffer of the specified device.

A communication cost of size $(n-1) * |T|$ is generated, where $p_1$ is the number of devices on which the input tensor is placed and $|T|$ is the size of the logical tensor T. Then, the data from the specified device is broadcast to device 0 and device 1, resulting in a communication overhead of size $p_2 * |T|$, where $p_2$ is the number of devices to place the output tensors. We use the communication primitive AllReduce to achieve it, which generates an overall transfer cost of size $(p_1 - 1 + p_2) * |T|$.

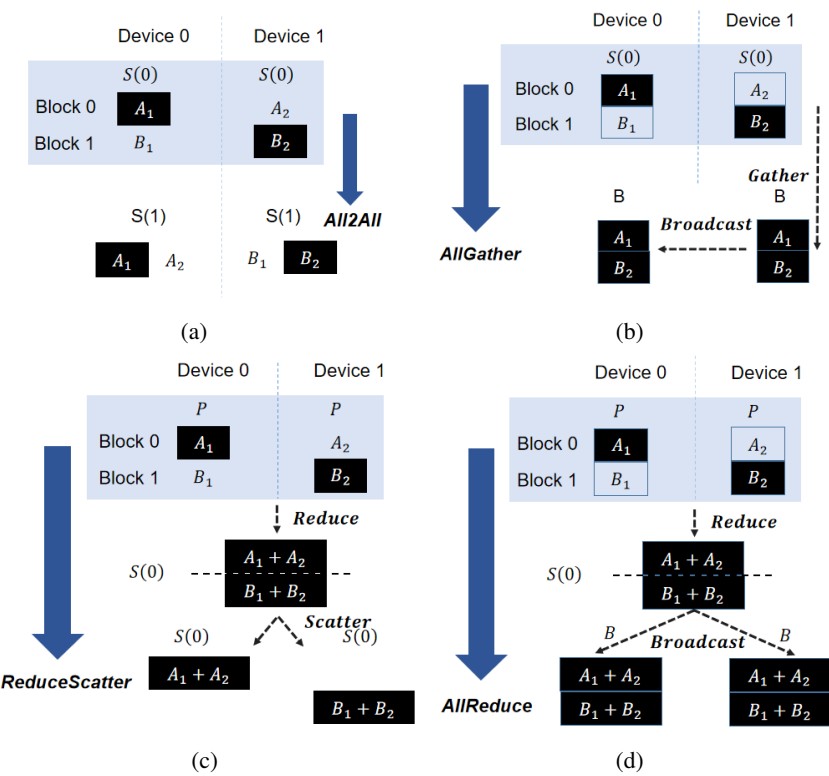

Figure 3: $S_i 2 S_j$ conversion(a), $S2B$ conversion(b), $P2S$ conversion(c), and $P2B$ conversion(d). Reduce operation is performed for two tensors with PartialReduce $P$ in both devices, where $P$ decides the tensor shape to match the shape of the global logical tensor.

## A.4 SUPPLEMENTARY EXPERIMENTS

**Large-Scalability**. This experiment is intended to observe the DistPar's implementation of the hyperscale face recognition model, insightface. The size of the throughput on the insightface model was experimented on different batch sizes and the number of categories. The server is configured with 8-card NVIDIA Tesla V100, FP32. Moreover, data parallelization with Broadcast distributed properties and model parallelization with S1 distributed properties are involved.

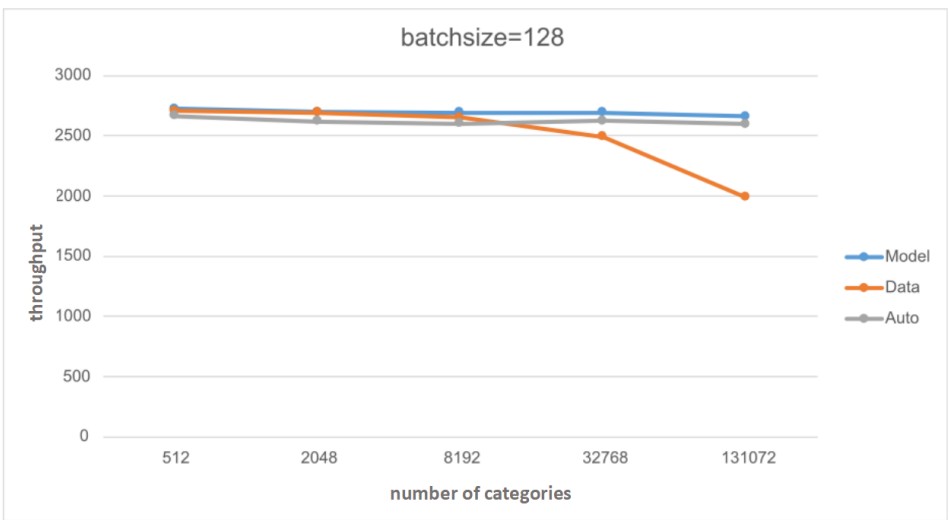

Figure 4: Performances of DistPar, data parallelization and model parallelization, with batch_size fixed to 128. As the batch size grows, DistPar is similar to model parallelism in performance. Nevertheless, it's worth noting that when batch size is 128, DistPar performs worse than model parallelism. However, by properly adjusting the ratio of computational cost to communication cost, DistPar could improve its performance which is comparable to model parallelism.