# OpenReview forum: "DISTPAR:TENSOR PARTITIONING FOR DISTRIBUTED NEURAL NETWORK COMPUTING"
_ICLR.cc/2024/Conference — ICLR 2024 Conference Withdrawn Submission_

### Official Review · Reviewer_Q7ud · 2023-10-30

**Soundness:** 1 poor
**Presentation:** 2 fair
**Contribution:** 2 fair
**Rating:** 1
**Confidence:** 4

**Summary:**

The paper presents DistPar, a distributed training framework for developing parallel models that achieves decent results in face recognition and language modeling tasks. DistPar uses tensor parallelism, expanding the search space for optimal parallel strategies. The framework also uses a cost function that balances computational and communication costs to select the best-performing parallel scheme. The authors analyze the system performance in view of throughput and speedup, conducting a comparative evaluation on the performance of DistPar's automatically selected parallelism strategy against data parallelism in PyTorch and TensorFlow frameworks. Overall, the paper presents a promising approach to parallel model development with potential applications in a wide range of fields.

**Strengths:**

This paper tackles a timely and important problem: given the rise of foundation models in the ML community, distributed training (especially model parallelism) has become a necessity to break the accelerator's memory wall. However, writing model parallel code requires a careful balance between communication and computation. Thus, such an auto-parallel framework will be very useful.

DistPar also presents state-of-the-art results: the authors demonstrate results better than DP in experiments, achieving 50% higher throughput in large-scale face recognition tasks and a 20% improvement in language modeling tasks compared with data parallelism provided by PyTorch.

**Weaknesses:**

# Concerns about the integrity of the evaluation.
1. Fig.2: The shapes of Fig. 2a and 2c seem identical for two vastly different models with different computation and communication patterns. Why is that the case? Furthermore, TensorFlow throughput is lower than PyTorch on 2a and 2c, but its speedup is higher on 2b and 2d. Why is it so?
2. It would be great to show an ablation study on performance improvement.

# Questionable cost model design.
1. In DistPar, the computational cost is modeled as the sum of elements of the input and output tensor regardless of the operator used. This design is questionable and uncomparable, as different operators (for instance, matmul v.s. element-wise operators) have very different FLOPs. Some ops can even have a significant input/output tensor without any computation done (eg: memcopy operation).

2. The cost model uses `beta` to somehow model the arithmetic intensity, and the tuning of `beta` was presented in the evaluation (Sec4.2). The authors claim that jointly considering computation+communication cost is better than considering computational cost alone. However, this is not supported in Fig.3, when computational cost alone performs very well (where `beta=max`). Does it mean that the sum of input/output tensor alone can model the performance of the DNN models? Why is the performance of DNN models consistent across a very large range of beta? Why is the unit of the y-axis in MB/sec?


# Poor writing
This submission contains some unjustified claims/details and grammar errors. For instance:
- "DistPar needs to dynamically derive intermediate transformation primitives." -> How is the automatic transformation performed?
- Sec 4.3 -> "The configured with 8 GPUs of NVIDIA Tesla V100, FP32".

# Lack of novelty
The submission presents 4 IRs: placement, broadcast, scatter, and partial reduce. However, similar ideas have been proposed previously [1,2,3]. It would be great to explain the difference between the new IRs and prior works, and justify why the new IRs are necessary.

---
Reference:
1. Alpa: Automating Inter- and Intra-Operator Parallelism for Distributed Deep Learning
2. OneFlow: Redesign the Distributed Deep Learning Framework from Scratch
3. ATP: Adaptive Tensor Parallelism for Foundation Models

**Questions:**

Please clarify the questions listed above. On top of that, I have additional questions below:

1) In Fig.4a, what is the auto-strategy selected, and why can it outperform both data and model parallel strategies? What is the model size (in the number of parameters) for Fig 4's x-axis?

2) Fig 5: What is the setup (#params, dimension of last three layers) for the model given? Can you visualize the best parallel strategy (RCR)? What insights can we draw from it? Is pytorch baseline pure data parallel strategy, and if so, does it incur an out-of-memory issue?

3) Sec 4.1: What was DistPar implemented on? What is the setup used?

---

### Official Review · Reviewer_pTBS · 2023-11-02

**Soundness:** 1 poor
**Presentation:** 2 fair
**Contribution:** 1 poor
**Rating:** 3
**Confidence:** 4

**Summary:**

This work, DistPar, proposes an unified global tensor for efficient tensor parallelism. Key concepts, design elements, and methodology are introduced. Comparision with TensorFlow DP and PyTorchDDP are made.

**Strengths:**

+. Willing to open-source code

**Weaknesses:**

-. **Limited Contribution**

> Our unified strategy, DistPar, introduces a set of tensor partitioning attributes aimed at instructing the allocation of global logical tensors to specific physical devices—referred to as physical tensors for simplicity. DistPar merges these devices into a coherent logical supercomputer, allowing developers to handle parallel training tasks on multiple devices as simply as a single device. This enhanced accessibility for individual users, so they can focus on more top-level design.

How is this work different from TensorFlow's DTensor, Colossal-AI's DTensor, and OneFlow's Global Tensor?

-. **Limited Comparision**

> DistPar reaches 50% higher throughput in large-scale face recognition tasks and
a 20% improvement in language modeling tasks compared with data parallelism
provided by PyTorch.

Why Tensor Parallelsim work is compared PyTorchDDP? For fairness, how about comparing with other Tensor Parallelism works like PyTorchFSDP, Alpa (its intra-op parallelism), Colossal-AI (its tensor parallelism part), DeepSpeed (its Automatic Tensor Parallelism), and Megatron-LM?

-. **Limited Evaluation**

> We conducted a comparative evaluation, analyzing ResNet-50 pre-trained on the ImageNet-
2012 dataset (Heigold et al., 2013) for image recognition and the BERT-Base model (Karpathy
et al., 2014) for query answering in natural language processing tasks.

Why only ResNet50 and BERTBase? They just fit a single GPU, and Tensor Parallelism is not a requirement.

How about other models like GPT and Llama that truly require Tensor Parallelism?

-. **Limited Benefit**

> Figure 2

It seems this work doesn't improve speed over TensorFlow DP and PyTorchDDP much.

-. **Unclear Method**

> Computational Cost in DistPar is simplified to the sum of the elements of the input and output
tensors corresponding to different parallelization strategies

Is this cost model too simple to find the best parallelization strategy?

> Communication Cost is defined as the total communications across multiple devices. In our implementation, communication cost is estimated using the conversion cost that results from the conversions of distributed properties.

What are the formulas and algorithms to model this communication cost?

**Questions:**

*. See above

---

### Official Review · Reviewer_ovib · 2023-11-03

**Soundness:** 3 good
**Presentation:** 2 fair
**Contribution:** 1 poor
**Rating:** 1
**Confidence:** 5

**Summary:**

This paper presents DistPar, a user-friendly framework for developing parallel models. It simplifies the process of writing programs for multiple devices by providing primitives and a cost function that automatically selects the best configurable parameters. The results demonstrate that DistPar can achieve a 50% increase in throughput for large-scale face recognition tasks and a 20% improvement in language modeling tasks compared to data parallelism.

**Strengths:**

- Extensive experiments to show the advantages of using DistPar
- A logical global view to ease the programming

**Weaknesses:**

- The paper lacks originality as most of its content, which includes the search space and search algorithm, has already been studied in [1, 2, 3, 4, 5]. Particularly, [1, 2] deal with a more complex version of the automatic search problem, which has a larger search space. Moreover, the concepts of logical view, distributed properties, and conversion have also been discussed in [3, 4, 5]. This paper fails to provide a comparison and fails to mention most of these highly related works.
- Furthermore, the experiments conducted are outdated and limited in scale. Most of the experiments are performed on small networks such as ResNet-50 and BERT with a small number of devices (<=32).

[1] Alpa: Automating Inter- and Intra-Operator Parallelism for Distributed Deep Learning, OSDI 22.
[2] Unity: Accelerating DNN Training Through Joint Optimization of Algebraic Transformations and Parallelization, OSDI 22.
[3] GShard: Scaling Giant Models with Conditional Computation and Automatic Sharding, ICLR 21.
[4] GSPMD: General and Scalable Parallelization for ML Computation Graphs, arXiv 21.
[5] OneFlow: Redesign the Distributed Deep Learning Framework from Scratch, arXiv 21.

**Questions:**

1. How does it compare to the related work mentioned above?
2. "Pytorch, its DistributedDataParallel interface is challenging to users, because it requires users to design the communicative module of parallelism strategies manually. ". This is not true. DistributedDataParallel and FSDP in Pytorch are single-line wrappers and do not require users to call all-reduce or other primitives manually.
3. How does it scale to large models like GPT models with hundreds of billions of parameters?